# Genetic Diversity, Evolutionary Dynamics, and Ongoing Spread of Pedilanthus Leaf Curl Virus

**DOI:** 10.3390/v15122358

**Published:** 2023-11-30

**Authors:** Zafar Iqbal, Muhammad Shafiq, Muhammad Naeem Sattar, Irfan Ali, Muhammad Khurshid, Umer Farooq, Muhammad Munir

**Affiliations:** 1Central Laboratories, King Faisal University, Al-Ahsa P.O. Box 55110, Saudi Arabia; mnsattar@kfu.edu.sa; 2Department of Biotechnology, University of Management and Technology, Sialkot Campus, Sialkot P.O. Box 51340, Pakistan; shafiq.4721@gmail.com; 3Centre of Agricultural Biochemistry and Biotechnology, University of Agriculture, Faisalabad P.O. Box 38000, Pakistan; irfan.cabb@uaf.edu.pk; 4School of Biochemistry and Biotechnology, University of the Punjab, Lahore P.O. Box 54590, Pakistan; khurshid.ibb@pu.edu.pk; 5Department of Biotechnology, University of Sialkot, Sialkot P.O. Box 51340, Pakistan; hu141985@gmail.com; 6Date Palm Research Center of Excellence, King Faisal University, Al-Ahsa P.O. Box 31982, Saudi Arabia; mmunir@kfu.edu.sa

**Keywords:** pedilanthus leaf curl virus, begomovirus, mutation, recombination, genetic diversity, phylogeny

## Abstract

Pedilanthus leaf curl virus (PeLCV) is a monopartite begomovirus (family *Geminiviridae*) discovered just a few decades ago. Since then, it has become a widely encountered virus, with reports from ca. 25 plant species across Pakistan and India, indicative of its notable evolutionary success. Viruses mutate at such a swift rate that their ecological and evolutionary behaviors are inextricably linked, and all of these behaviors are imprinted on their genomes as genetic diversity. So, all these imprints can be mapped by computational methods. This study was designed to map the sequence variation dynamics, genetic heterogeneity, regional diversity, phylogeny, and recombination events imprinted on the PeLCV genome. Phylogenetic and network analysis grouped the full-length genome sequences of 52 PeLCV isolates into 7 major clades, displaying some regional delineation but lacking host-specific demarcation. The progenitor of PeLCV was found to have originated in Multan, Pakistan, in 1977, from where it spread concurrently to India and various regions of Pakistan. A high proportion of recombination events, distributed unevenly throughout the genome and involving both inter- and intraspecies recombinants, were inferred. The findings of this study highlight that the PeLCV population is expanding under a high degree of genetic diversity (π = 0.073%), a high rate of mean nucleotide substitution (1.54 × 10^−3^), demographic selection, and a high rate of recombination. This sets PeLCV apart as a distinctive begomovirus among other begomoviruses. These factors could further exacerbate the PeLCV divergence and adaptation to new hosts. The insights of this study that pinpoint the emergence of PeLCV are outlined.

## 1. Introduction

In the intricate web of plant–pathogen interactions, viruses pose significant challenges to global agriculture by inflicting diseases that have the potential to devastate crops. Among this vast array of viral pathogens, the pedilanthus leaf curl virus (PeLCV) emerges as a notable member of the begomovirus family, exerting its influence on plant species across diverse agro-ecological regions of Pakistan and India. PeLCV, a typical Old World (OW) monopartite begomovirus (family *Geminiviridae*), harbors a ~2.8 kb single-stranded DNA genome encapsidated in a peculiar ‘geminate’ particle. It is transmitted by whiteflies (*Bemisia tabaci*), making it an adept plant-to-plant pathogen with far-reaching implications for agriculture. Globally, the whitefly species complex, particularly *B. tabaci* within the *Aleyrodidae* family, exhibits substantial diversity, comprising approximately 1500 cryptic species [1]. These species vary in terms of biology, host specificity, and geographic distribution, rendering the understanding and management of the whitefly complex increasingly intricate. A study by Fiallo-Olivé et al. [2] highlights this diversity, outlining various biotypes like A, B (Middle East-Asia Minor 1-MEAM1), Q (Mediterranean-MED), and other Asian variants. Particularly, in Pakistan and India, the B biotypes, including MEAM1, Asia I (biotypes H, M, NA), Asia II-1 (biotypes K, P, ZHJ2), and Asia II-5, are more prevalent [3,4]. The emergence of new combinations of begomovirus or the introduction of ed begomovirus species to new geographic niches has been facilitated by a combination of factors, including the global proliferation of the whitefly vector, the extensive presence of non-cultivated host plants, and human activities [5,6].

Initially identified in *Pedilanthus tithymaloides*, an ornamental shrub, in 2009 [7], PeLCV has since been shown to affect more than 25 different plant species, including carrot [8], *Chenopodium album* [9], chili pepper [10], *Petunia atkinsiana* [11], radish [12], soybean [13], *Sesbania bispinosa* [14], spinach [15], turnip [16], fenugreek, and Jasmine [17]. In most instances, PeLCV was found to be associated with tobacco leaf curl betasatellite (TbLCuB). However, recent findings have shown its presence alongside other betasatellites. In Pakistan, PeLCV has been reported with three distinct betasatellites: TbLCuB, cotton leaf curl Multan betasatellite, and digera yellow vein betasatellite (DiYVB) [11,16]. Likewise, in India, it has been reported with three different betasatellites: DiYVB, tobacco leaf curl chili betasatellite, and tomato leaf curl Patna betasatellite [8,18]. Additionally, alphasatellite, another type of DNA satellite, has also been identified in association with PeLCV. In Pakistan, one alphasatellite, *ageratum conyzoides* symptomless alphasatellite, has been documented in PeLCV infections in *Cassia occidentalis* and fenugreek plants [17,19].

PeLCV encodes six open reading frames (ORFs) in both virion-sense and complementary-sense orientations, separated by a non-coding intergenic region spanning promoter elements and the origin of replication, encompassing a hairpin loop nonanucleotide (TAATATTAC) sequence [20]. Two ORFs, V1 (coat protein [CP]) and V2 (pre-coat protein), are encoded on the virion-sense strand. On the other hand, four ORFs, C1 (replication-associated protein (Rep)), C2 (transcriptional activator protein (TrAP)), C3 (replication-enhancer protein (REn), and C4, are encoded on the complementary-sense strand. All the functions ascribed to these ORFs have been thoroughly demonstrated [20,21]. Importantly, the mutational analysis of four PeLCV-encoded ORFs, V1, V2, C2, and C4, has revealed their indispensable role in ensuring successful and symptomatic viral infection [22].

Although PeLCV was identified just a couple of decades ago, it remains sporadic. However, in recent years, it has increasingly been identified from a diverse range of plant species, indicating its remarkable adaptability, spread, and undeniable evolutionary success. This has alarmed researchers, prompting them to put substantial research focus on it. Another notion that renders PeLCV particularly intriguing is its recurrent identification in weed plants. These weedy hosts serve as reservoirs and melting pots for viral dissemination, recombination, genetic exchange, and satellite capture [23,24,25]. The combination effect of all such genetic factors in the viral genome and selection pressure imposed by the host reconciles genomic variation, evolution, and adaptation of viruses to different hosts [26,27]. The genetic variation level of different geminiviruses is remarkably high and comparable to several RNA viruses [28,29,30,31] and may be an important arsenal for quick adaptation to a changing environment. Recombination is another notable driver for swift evolution. Several viral species, including geminiviruses such as Maize streak virus in Africa, tomato yellow leaf curl, cotton leaf curl, and cassava mosaic diseases-associated begomoviruses in Europe, Asia, and Africa seem to have evolved largely by recombination [30,32,33,34,35,36,37,38].

This study aimed to identify the geographic dispersion and decipher the insights that were shaping the genetic stature of the PeLCV population for swift evolution. The overarching objectives of this study were to undertake a detailed analysis of PeLCV phylogeography, decipher the evolutionary lineage and selection pressure, and explore genetic diversity indices. The significance of the findings is thoroughly discussed.

## 2. Materials and Methods

### 2.1. Sequence Retrieval and Multiple Sequence Alignment

A total of 52 full-length genome sequences of PeLCV were retrieved from NCBI GenBank (https://www.ncbi.nlm.nih.gov/), accessed on 25 February 2023 (Table 1). Subsequently, all the retrieved sequences were divided into six subsets to yield each ORF datum. Furthermore, all the retrieved sequences were further sub-grouped into country-wise data. So, a total of nine datasets were generated, comprising the entire PeLCV population (PeLCV-all), PeLCV isolates reported from Pakistan (PeLCV-Pak; 45 isolates) and India (PeLCV-Ind; 7 isolates), and six ORFs datasets. All the datasets were subjected to multiple sequence alignment (MSA) in MEGA11 using the Muscle algorithm [39,40], and MSA files were reviewed and adjusted if needed.

### 2.2. Time and Tree Root Estimation

TempEst is an invaluable tool used in evaluating the temporal signal and ‘clocklikeness’ within molecular phylogenies [41]. In our study, TempEst was harnessed to appraise both the contemporaneous and dated-tip trees of PeLCV, providing a critical examination of the adherence to molecular clock assumptions. Additionally, it was used to assess the tree root to gain insights into evolutionary timelines and phylogenetic investigations.

### 2.3. Phylogeny and Phylogeography

The full-length sequences of PeLCV-all were used to infer the best-fit nucleotide substitution model and then phylogenetic relationship and evolutionary relatedness using the maximum likelihood (ML) method with 1000 bootstrap replicates. The resulting tree was visualized and edited in Interactive Tree of Life (iTOL v6.5; https://itol.embl.de/#, accessed on 3 March 2023) [42].

To infer phylogeographic history of PeLCV, a time-scaled phylogeny based on the Bayesian framework (BSSVS) approach was opted by implementing character mapping in Bayesian software v.1.0 [43]. The PeLCV-all dataset was subjected to the Bayesian Evolutionary Analysis Utility (BEAUti2) program to set the attributes, such as tree priors, strict and relaxed (uncorrelated lognormal) molecular clocks, coalescent constant demographic models, and to generate the Bayesian Evolutionary Analysis Sampling Tree (BEAST v.2.7.4; [44] control file. Output of BEAST was visualized in Tracer v.1.7.2 [45]. To determine statistical uncertainty, different Markov chain Monte Carlo (MCMC) chain lengths were tried to achieve the best-fit molecular clock, effective sample sizes (ESS), and highest probability density (HPD ≥ 95%) interval. After visualizing and achieving all the desired traits, the BEAST-generated tree file was opened in TreeAnnotator v.1.10.5 [46] to yield a maximum clade credibility (MCC) tree, which was then visualized and annotated in FigTree v1.4.4 (http://tree.bio.ed.ac.uk/software/figtree/ accessed on 18 Auguust 2023). Finally, Spatial Phylogenetic Reconstruction of Evolutionary Dynamics using Data-Driven Documents (SpreaD3; [47]) was used to analyze and visualize PeLCV evolutionary phylogeny.

### 2.4. PeLCV Population Structure Assay

Genetic diversity indices (GDI), including nucleotide diversity (π), divergence, and other nucleotide diversity-related parameters of all the generated PeLCV datasets, were computed using DnaSP v.6.12 [48], as described earlier [31,49]. The total nucleotide diversity of each dataset was inferred using a 100-nucleotide sliding window with a step size of 25 nucleotides. The 95% bootstrap confidence intervals were used to estimate statistically significant differences in mean nucleotide diversity across all datasets.

The neutrality tests, Tajima’s D (TD) and Fu and Li’s D (FLD), represent the difference between two measures of genetic diversity, and both were inferred using DnaSP to assess the selection pressure on the PeLCV population and their individual ORFs.

### 2.5. Network Analysis

For network analysis, aligned CP ORF dataset of 52 PeLCV isolates was used to infer the median-joining (MJ) network of PeLCV in popART v.1.7 [50]. DnaSP was used to infer geographical and host-wise distribution, and then nexus output file was subjected to popART to infer MJ tree using zero epsilon. Sequences with 100% sequence homology were designated as one sequence.

### 2.6. Estimation of Nucleotide Substitution Rate

Nucleotide substitution.site^−1^.year^−1^ (NSSY) and mutations in PeLCV-all, PeLCV-Pak, PeLCV-Ind, and all the ORFs datasets were calculated using the MCMC approach in BEAST (v.1.10.5) with 1 × 10^8^ chain length [46]. All the PeLCV datasets were examined through both strict molecular and relaxed molecular (uncorrelated lognormal) clocks. BEAST output was analyzed in Tracer to assess the best-fit clock, mutation rate at three different codon positions of ORFs, and ESS (≥200).

### 2.7. Selection Pressure Analysis

To determine the selection pressure (negative or positive) on the PeLCV-encoded ORFs, two approaches were employed. First, the ratio of non-synonymous (dN) to synonymous (dS) substitutions was inferred in MEGA 11. Second, an online web-based server, Datamonkey ([51]; www.datamonkey.org; accessed on 28 February 2023), was exploited using Fast, Unconstrained Bayesian AppRoximation (FUBAR) and single-likelihood ancestor counting (SLAC) methods.

### 2.8. Recombination Analysis

Two distinct approaches were used to detect potential recombination events in PeLCV-all, PeLCV-Pak, and PeLCV-Ind. The first, Genetic Algorithm for Recombination Detection (GARD), an online tool (http://datamonkey.org/; accessed on 28 February 2023), and the second, Recombination Detection Program (RDP v.5.5) [52], was used.

GARD analysis was executed with rate classes of 4, site-to-site variation type of Beta-Gamma, and normal run mode. In RDP, seven different algorithms (BOOTSCAN, CHIMAERA, GENECONV, RDP, MAXCHI, SISCAN, and 3SEQ) were implemented with default detection thresholds at *p*-value of 0.05. Only those recombination events were considered credible that were detected by at least 4 different algorithms.

## 3. Results

### 3.1. Evolutionary Time Estimation, Phylogeny, and Network Analysis

In the PeLCV TempEst analysis, a neighbor-joining phylogenetic tree, constructed in MEGA11 without a molecular clock, was used. This tree was instrumental in implementing linear regression models to evaluate the root-to-tip genetic distance concerning the sampling time for each tip of PeLCV. The analysis yielded a rate of genetic evolution of 1.54 × 10^−3^. Furthermore, the point at which this regression intersected the time axis occurred in 1977.12 (Appendix A).

Phylogenetic analysis grouped the 52 PeLCV sequences into seven major clades, displaying some regional delineation but lacking any host-specific demarcation (Figure 1). Similarly, MJ network analysis grouped 52 PeLCV isolates into seven distinct clades (I–VII), displaying some regional delineation but lacking any host-specific demarcation (Figure 2). Clade I included 12 PeLCV isolates (8 from Pakistan and 4 from India), reported from vegetables, papaya, weeds, and ornamental plants. Clade II, on the other hand, consisted of only two isolates from India, identified from whitefly and papaya. Clade III contained seven PeLCV isolates reported from weeds, ornamental plants, and vegetables in Pakistan. Similarly, clades IV and V each contained eight isolates from Pakistan, reported from papaya, weeds, ornamental plants, and vegetables. Furthermore, all of the 12 isolates in clade VII were from Pakistan, with the exception of one isolate (OM144969) from India. Notably, one isolate, MK158209, reported from *Albizia lebbeck* from Pakistan, formed a monophyletic clade in both analyses.

In MJ network analysis, single-deletion events were coded between adjacent nucleotides, and the data were analyzed with epsilon parameters ranging from 0 to 5, yielding consistent results. Genealogical network analysis identified seven distinct population groups (I–VII) without host-specific demarcation (Figure 2). Notably, certain accessions are displayed.

There were some identical sequences, such as AM948961 with MG764701-MG764703 and MG764705, and KX671561 with KX671562 and KX711622. Furthermore, two closely related groups, OK236813 (from papaya) and ON054966 (from whitefly) in India, shared a few mutation points. Interestingly, PeLCV isolates from *Glycine max* shared three distinct clades (Clade V, VI, and VII), one grouping with PeLCV from weeds (MN885484 and MN885485), the second with PeLCV from *G. max* and weeds (MN885482 and MN885483, KX671561, and KX671563), and the third with PeLCV reported from vegetables. Except for group III, PeLCV isolates from various vegetables like spinach, tomato, turnip, and radish were evenly distributed among the other groups (Figure 2). Overall, in the MJ tree, all the clades were connected by small black circles, representing several missing sequences of the PeLCV isolates.

### 3.2. Genetic Diversity Indices

The dynamics of sequence variation were assessed in PeLCV-all, PeLCV-Pak, and PeLCV-Ind, along with all the ORF datasets. Overall, PeLCV-all was found to be highly dynamic and harbored the highest values of different genetic diversity attributes, followed by PeLCV-Pak and PeLCV-Ind populations. Among PeLCV-encoded ORFs, Rep had the highest values for most of the investigated genetic diversity attributes, while V2 had the lowest (Table 1).

PeLCV-all had the highest number of polymorphic sites (S) and total mutation (Eta), 851 and 1057, respectively (Table 1). PeLCV-Pak had higher S (794) and Eta (949) values than PeLCV-Ind, with S and Eta values of 609 and 684, respectively. Among the ORFs, Rep had the highest S (351) and Eta (433), followed by CP (S = 219; Eta = 266). Interestingly, V2 had the lowest S (92) and Eta (115) values. The highest number of InDels was observed in PeLCV-Pak, PeLCV-all, PeLCV-Ind, TrAP, REn, and Rep: 152, 86, 62, 24, 21, and 18, respectively (Table 1). Notably, no InDels were found in the C4 ORF. Among all analyzed datasets, a total of 272 pairwise nucleotide differences (K) were found in PeLCV-Ind: 175 in PeLCV-Pak, 172.08 in PeLCV-all, 59.56 in CP, and the fewest in REn (14.97) (Table 1).

The K estimation revealed that PeLCV-all and PeLCV-Pak have the same level of genetic differentiation between them. Nevertheless, PeLCV-Ind has a higher level of genetic differentiation than PeLCV-all and PeLCV-Pak (Table 1).

The average nucleotide diversity (π) for all the datasets showed that PeLCV-Ind had the highest π value (0.10%), followed by PeLCV-all (0.073%) and PeLCV-Pak (0.070%) (Figure 3). Among the ORFs, CP and Rep had the highest π value of 0.077%, while TrAP had the lowest, 0.039% (Figure 3). Moreover, Ѳw was substantially higher in PeLCV-Ind (0.092) than in PeLCV-all (0.080) and PeLCV-Pak (0.073), revealing the presence of a greater number of polymorphic sites. On the other hand, Rep and TrAP had substantially more segregation sites of 0.089 and 0.084, respectively. Meanwhile, the lowest number of average segregation sites was 0.063 in CP and V2 (Figure 3).

Furthermore, the total nucleotide diversity across the nucleotide positions of all the PeLCV datasets was also conducted (Appendix A). The analysis revealed that in the PeLCV-all, PeLV-Pak, and PeLCV-Ind datasets, the nucleotide diversity rates generally remained below 0.2%. However, there was an exception of N-ter of Rep of PeLCV-Ind, which had the greatest nucleotide diversity. Notably, the middle and C-ter of CP in all three populations showed high nucleotide diversity. Likewise, the N-ter of REn demonstrated the highest nucleotide diversity.

### 3.3. Neutrality Indices

In all of the datasets examined, the neutrality indices calculated by TD and FLD tests were negative (Figure 4; Table 1). The TD values were highly negative, except for V2 and C4. PeLCV-all (−2.01), PeLCV-Pak (−1.70), CP (−1.46), and TrAP (−1.46); all had high negative TD values, while V2 had the lowest (−0.24) value (Figure 4).

Likewise, the FLD values were significantly negative, except for CP, PeLCV-Ind, and C4. The highest negative FLD values were inferred for TrAP (−3.67), REn (−2.98), C4 (−1.98), and Rep (−1.87). Notably, PeLCV-Ind had the lowest FLD (−0.066) value, indicating that natural selection has a minor role in the diversification of PeLCV-Ind (Figure 4).

### 3.4. Estimation of Selection Pressure

FUBAR, SLAC, and dN/dS ratios were used to assess negative selection pressure on each PeLCV-encoded ORF (Table 2). According to FUBAR results, fewer ORF sites showed evidence of positive selection; C4 and V2 had the most positive sites, nine and eight, respectively. Notably, SLAC data found just a few sites with positive selection in TrAP, while none were found in any of the other ORFs.

On the other hand, more sites in the ORFs with evidence of negative selection were observed. FUBAR and SLAC detected 112 and 46 sites with negative selection in Rep, followed by V2 (110 and 8) and CP (99 and 50) (Table 2).

The average dN/dS ratio for all the ORFs, except V2 (4.86) and Rep (1.36), was recorded as less than one. The lowest ratios were inferred for CP (0.23) and REn (0.31), demonstrating that the genomic variations are mainly due to natural or purifying selection on these ORFs. However, it is important to note that V2 is undergoing diversification under significantly robust positive selection pressure, whereas Rep is experiencing comparatively weaker positive or adaptive selection pressure.

### 3.5. Evolutionary Rate Estimation

The rate estimates and the mean of nucleotide substitutions were assessed at the 95% HPD interval with both strict- and relaxed uncorrelated molecular clocks. The rate estimates span different ranges of NSSY substitution in all the nine datasets, ranging from 2.09 × 10^−2^ to 8.27 × 10^−4^ substitution.site^−1^.year^−1^ at 95% HPD interval with a strict molecular clock (Table 3). For PeLCV-all, PeLCV-Pak, and PeLCV-Ind, the rates were highest in PeLCV-all (8.19–8.27 × 10^−4^ sub.site^−1^.year^−1^), followed by PeLCV-Ind (8.16–8.26 × 10^−4^) and PeLCV-Pak (1.43–1.44 × 10^−4^). Among the PeLCV-encoded ORFs, TrAP (3.21–3.7127 × 10^−4^) had the highest and V2 (2.09–2.76 × 10^−2^) had the lowest sub.site^−1^.year^−1^ (Table 3).

The effects of nucleotide substitution on the three-codon positions (CoP) of all PeLCV-encoded ORFs using both strict and relaxed molecular clocks were also assessed. Our results demonstrated that only V2 had a higher mutation rate at CoP1, while C4 had a higher mutation rate at CoP2, and the other ORFs had a higher mutation rate at CoP3 (Table 4).

### 3.6. Recombination Analysis

The RDP analysis unveiled that the PeLCV population was evolving through recombination. In most of the PeLCV-all genomes, multiple recombination events were detected (Appendix A). Among the 52 PeLCV-all isolates, 47 harbored 85 credible recombination events. Additionally, several recombination breakpoints were also inferred, spanning different regions of the genes. Notably, the hot spot regions of recombination breakpoints were identified at the C-ter of V2; the N- and C-ter of CP; and the N-, middle, and C-ter of Rep. In the PeLCV-Pak dataset, 39 of 45 sequences contained credible recombination events, and KX710160, MN566098, OM993555, and OM993556 had three recombination events in their genomes (Appendix A). Among the seven PeLCV-Ind isolates, five had recombination events, and JN807764, JQ12790, and KX168428 harbored three recombination events in their genomes (Appendix A).

GARD analysis inferred seven breakpoints in the PeLCV-all dataset at ∆C-AIC (vs. null model) = 3330.64; ∆C-AIC (vs. the single tree multiple partition) = 1.76 × 10^3^. Most of the recombination breakpoints were detected around nucleotide positions of 400, 1100, 1650, 1950, 2300, 2500, and 2600 (Figure 5A). Likewise, GARD analysis inferred seven breakpoints in PeLCV-Pak isolates at ∆C-AIC (vs. null model) = 2899.45; ∆C-AIC (vs. the single tree multiple partition) = 2.01 × 10^6^. Most of the recombination breakpoints were detected around nucleotide positions of 400, 1100, 1700, 1950, 2400, 2500, and 2600 (Figure 5B). In PeLCV-Ind isolates, GARD analysis inferred 12 breakpoints at ∆C-AIC (vs. null model) = 635.53; ∆C-AIC (vs. the single tree multiple partition) = 9.71 × 10^2^. Most of the breakpoints were detected around nucleotide positions of 100, 420, 600, 700, 1100, 1150, 1270, 1350, 1650, 1900, 2050, 2425, and 2600 (Figure 5C).

### 3.7. Time-Scaled Phylogeography of PeLCV

To trace the dissemination of PeLCV from its origin to various regions, we employed discrete trait models (in the BEAST program), utilizing empirical phylogenetic trees comprising 52 full-length sequences of PeLCV isolates. Figure 6 illustrates the MCC tree of PeLCV, where branches are color-coded based on the location at the tips. The sequences within each clade encompass host information, sample location, and chronological data. Notably, the time-scaled tree shared several common features with the ML phylogenetic tree and network analysis.

The time-scaled tree delineates seven major clades, characterized by the assortment of multiple PeLCV sequences, indicative of extensive genome reassortment. TempEST results demonstrated that the PeLCV progenitor evolved during 1977 (Appendix A). The MCC tree highlights the earliest genetic divergence occurring in 1982, with a subsequent bifurcation of one clade (upper segment of the tree) in 1993, showcasing the highest genetic diversity within the PeLCV population. In contrast, the other clade (lower segment of the tree) maintained a linear progression until 2015, displaying the lowest genetic diversity in the PeLCV population.

The time-scaled phylogeny elucidated that the PeLCV progenitor originated in Multan, Pakistan, in 1977, where it remained localized until 1980. Thereafter, it began to disseminate to other regions of Pakistan and India. The earliest discernible PeLCV dispersal event involved an unknown PeLCV strain moving from Multan, Pakistan, to New Delhi, India, between 1980 and 1990. Afterward, this was followed by three more movements in the 1990s, including a notable return from New Delhi, India, to Lahore, Pakistan. In 2000, PeLCV reached Rahim Yar Khan and Faisalabad, Pakistan, while it started spreading to Gujarat, India. A significant proliferation was observed in 2010, as PeLCV disseminated to Nawabshah, Pakistan, and Pusa and Lucknow, India. Further movements took place during and after 2020: it dispersed to Lahore, Sheikhupura, Dera Gazi Khan, Kohat, and Islamabad in Pakistan, as well as Gujarat in India.

Collectively, these data indicated that Multan was an epicenter of PeLCV origin. Our analysis revealed four notable long-distance movements of PeLCV, three of which were eastward movements: from Multan to New Delhi, Multan to Luckhnow, and Multan to Gujarat. The fourth movement occurred westward, from Luckhnow, India, to Kohat, Pakistan (Figure 6).

## 4. Discussion

Viruses are evolutionary beasts that evolve at an unprecedented rate, and they mutate so swiftly that their ecological and evolutionary lineages are inextricably linked. Their evolutionary footprints are imprinted as the genetic diversity on the genomes of the organisms, and these fingerprints can be traced through empirical analyses of evolutionary dynamics. Begomoviruses also exhibit a remarkably high level of diversity and a high propensity for fast evolution, which is comparable to several RNA viruses, rendering them potent pathogens. This study delves into the genetic diversity and evolutionary dynamics of all the PeLCV isolates currently circulating in Pakistan and India.

While many begomoviruses have a limited host range, PeLCV exhibits an exceptionally broad host range that makes it an intriguing subject for pathogenic investigation. PeLCV has been identified from several weed and perennial plant species [7], which acts as reservoirs for viruses and facilitates their dissemination, genome recombination, and satellite capture [49,53]. Host plants play a crucial role in the evolution of viruses through factors such as variable resistance or susceptibility levels. Additionally, the genetic diversity of host plants shapes the viral population dynamics and favors the selection of strains adapted to specific genetic environments. Viruses also exhibit preferences for certain plant species, each occupying distinct ecological niches, influencing viral replication and transmission through insect vectors. Consequently, the range of host plants a virus encounters shapes the selective pressures it faces, influencing its genetic diversity and adaptation process. Conversely, different begomoviruses exhibit varying transmission efficiencies by different cryptic whitefly species [54]. For instance, the efficient transmission of cotton leaf curl Multan virus with its cognate betasatellite in the Indo-Pak region has been observed by the cryptic species Asia II-1 compared to other whitefly biotypes [55,56]. PeLCV is prevalent in the same region, so speculatively, this biotype might also facilitate the transmission of PeLCV.

In phylogenetic, network, and time-scaled phylogeny analyses, all the PeLCV isolates coalesced into seven distinct clades, displaying regional delineation but lacking any host-specific demarcation. This suggests that PeLCV isolates are not constrained by host species in their evolutionary pathways. Notably, clades III, IV, V, and VI predominantly encompassed PeLCV isolates reported from Pakistan, while Indian isolates were clustered in clades I and II alongside with Pakistani isolates. These findings align with previous research, as earlier phylogenetic analyses of PeLCV had revealed the existence of discrete clusters marked by regional distinctions [11,16,17]. Nonetheless, these studies identified different numbers of clusters of PeLCV isolates. For example, Shakir et al. (2018) identified four major clusters, encompassing Cluster I and III, attributed to Pakistani isolates; Cluster II, representing Indian isolates; and Cluster IV, representing a distinct group of Indian and Pakistani isolates. The disparity in cluster number observed in these studies can be attributed to the small sample size. Our study encompassed a comprehensive dataset, incorporating all available PeLCV isolates, thus rendering it more comprehensive in scope. Additionally, time-scaled phylogeny data supported the notion that the progenitor of PeLCV originated in Multan, Pakistan, around 1977. This finding aligns with prior speculations regarding its Pakistani origin and subsequent spread to various agro-ecological zones in India [11,17].

The PeLCV population, regardless of its geographic origin, exhibited a high level of GDIs. The PeLCV-Pak dataset demonstrated higher InDels and segregating sites than PeLCV-Ind, whereas the PeLCV-Ind dataset displayed higher genetic variations (π and Ѳw) than PeLCV-Pak. These findings underscore the substantial sequence divergence and the higher frequency of unique mutations within the PeLCV population, emphasizing their roles in the emergence of PeLCV. Our results corroborated a study on chili leaf curl virus (ChiLCV) that highlighted similar diversity patterns [57]. Notably, papaya leaf curl virus (PaLCuV) exhibited low sequence divergence but a high frequency of unique mutations [58]. In contrast, euphorbia yellow mosaic virus and tomato leaf curl Palampur virus (ToLCPalV) had extremely low sequence divergence and a low frequency of unique mutations [31,59]. Our results on PeLCV-encoded ORFs unveiled different GDIs for different ORFs, pinpointing the non-random distribution of nucleotide diversity and genetic mutations along their sequences. Overall, Rep exhibited the highest values for different GDIs, while V2 displayed the lowest. Previous research findings concur that, typically, the Rep genes of begomoviruses have more nucleotide variability [30]. Additionally, our results align with the patterns observed in prior research on ToLCPalV, PaLCuV, and ChiLCV, where non-random distributions of nucleotide diversity and genetic mutations within their respective ORFs were reported, thereby corroborating the observed pattern [31,57,59]. The presence of uneven genetic variation distributions in PeLCV demonstrated that different factors affect genetic variability to varying degrees and at random locations. Such non-random distribution in various begomoviruses has previously been shown to play a role in genetic variability [58,60,61]. Furthermore, inter-population comparison (K) showed that PeLCV-all and PeLCV-Pak populations had a comparable level of genetic divergence, while the PeLCV-Ind population exhibited notably higher levels of genetic differentiation. Likewise, all PeLCV-encoded ORFs have low K values, reiterating the low level of genetic differentiation in them.

The neutrality indices across all the PeLCV datasets exhibited negative values, implying that PeLCV populations have more polymorphic sites or a high prevalence of low-frequency alleles that play a role in diversification [62]. Further, the results suggested that the PeLCV population is diversifying under purifying selection and population expansion, or they have experienced recent expansion rather than a neutral selection [58,61]. Nevertheless, the combined presence of TD and FLD negative values across PeLCV-all, PeLCV-Pak, PeLCV-Ind, and PeLCV-encoded ORFs signify the conserved nature of their genes. Such results are likely to occur in populations where the majority of observable nucleotide segregation is transient and is eventually flushed away by purifying selection [57]. Likewise, diverse dN/dS ratios within different PeLCV-encoded ORFs revealed the presence of diversifying selection. Specifically, the higher dN/dS ratio noted in V2 and Rep indicated that both these genes are evolving under purifying and strong negative selection. This indicates variable purifying selection among these ORFs, while more negatively selected sites lead to lower dN/dS ratios. Similar findings have previously been demonstrated in different viruses [63] and different begomoviruses—for instance, ChiLCV [57] and ToLCPalV [31].

PeLCV-all, PeLCV-Pak, and PeLCV-Ind datasets had notable NSSY rates, quantified at 8.22 × 10^−4^, 1.44 × 10^−4^, and 8.20 × 10^−4^, respectively, surpassing EACMV [64], ChiLCV [57], ToLCPalV DNA-A [31], TYLCV [65], and PaLCV [58]. It is pertinent to note that several geminiviruses have a high NSSY rate (of the order of 10^−4^), comparable to many RNA viruses [29]. Although we used both strict and relaxed molecular clocks to infer the NSSY rate, the strict molecular clock yielded more promising results. This outcome diverged from a previous study that favored the relaxed molecular clock [58].

Recombination analysis revealed dominant recombinants in the PeLCV population. The presence of unknown recombinant parents alongside interspecies parents suggested the occurrence of both inter- and intraspecies recombination events. Additionally, GARD analysis inferred several breakpoints in the PeLCV-all, PeLCV-Pak, and PeLCV-Ind datasets, with multiple breakpoints clustering in similar regions considered as ‘hot spots’ for recombination in PeLCV [66]. Recombination among all PeLCV datasets provided clues about the high rate of evolution, rapid multiplication, and host range expansion [23,34,67,68]. The rolling circle replication mechanism in begomovirus populations mechanistically predisposes them to frequent recombination events at non-random sites [69,70].

The findings of this study have several implications for the evolution of PeLCV. The high genetic diversity, rapid nucleotide substitution rates, and ongoing recombination events suggest that PeLCV is a dynamic and evolving virus. This adaptability may allow PeLCV to overcome host resistance mechanisms and expand its host range. The evolutionary dynamics of PeLCV are likely influenced by a combination of genetic and ecological factors.

## 5. Conclusions

The PeLCV population is expanding under demographic selection, pinpointing that its genetic variants have higher fitness in growing populations, leading to their increased prevalence. Additionally, it is plausible to consider that PeLCV identification from diverse plant species might serve as an additional evolutionary factor, as viruses often adapt and evolve in response to the selective pressures exerted by different host plants. Furthermore, the PeLCV population in Pakistan and India has a higher degree of intra- and inter-specific recombination. It is thus imperative that the PeLCV population is shaping up by all the possible means of evolutionary forces, like mutation, recombination, natural and positive selection, and genetic drift. Thus, PeLCV is a unique begomovirus from other commonly occurring begomoviruses, highlighting the complexity of the PeLCV population.

## Figures and Tables

**Figure 1 viruses-15-02358-f001:**
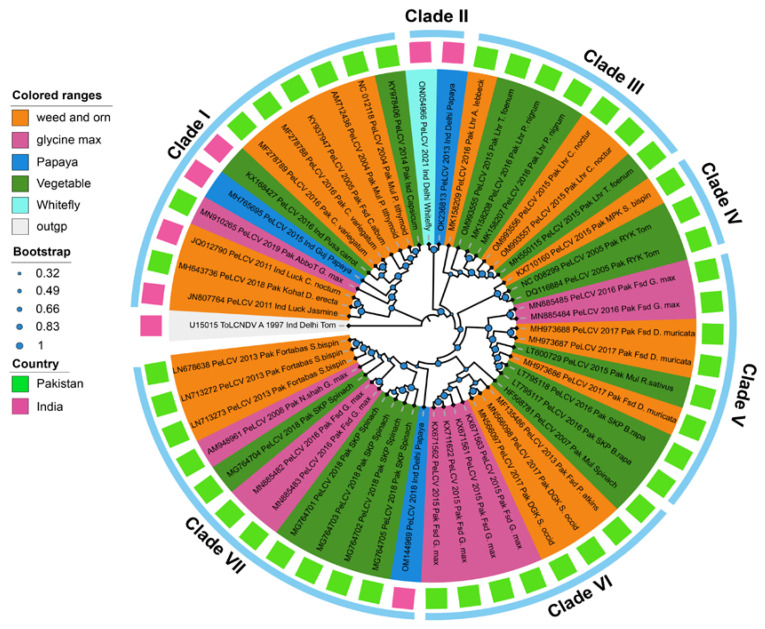
The phylogenetic dendrogram based on the complete nucleotide sequences of 52 PeLCV isolates from Pakistan and India. The phylogenetic tree was constructed with the Maximum-Likelihood (ML) algorithm in MEGA11 software using tomato leaf curl New Delhi virus (ToLCNDV; U15015) as an outlier. The inner ring colors indicate the host, and outer ring colors depict the country of origin of PeLCV isolates. The analysis was performed with 1000 replicates. All the related isolates used to construct phylogenetic tree were retrieved from NCBI database and are represented with their respective accession numbers.

**Figure 2 viruses-15-02358-f002:**
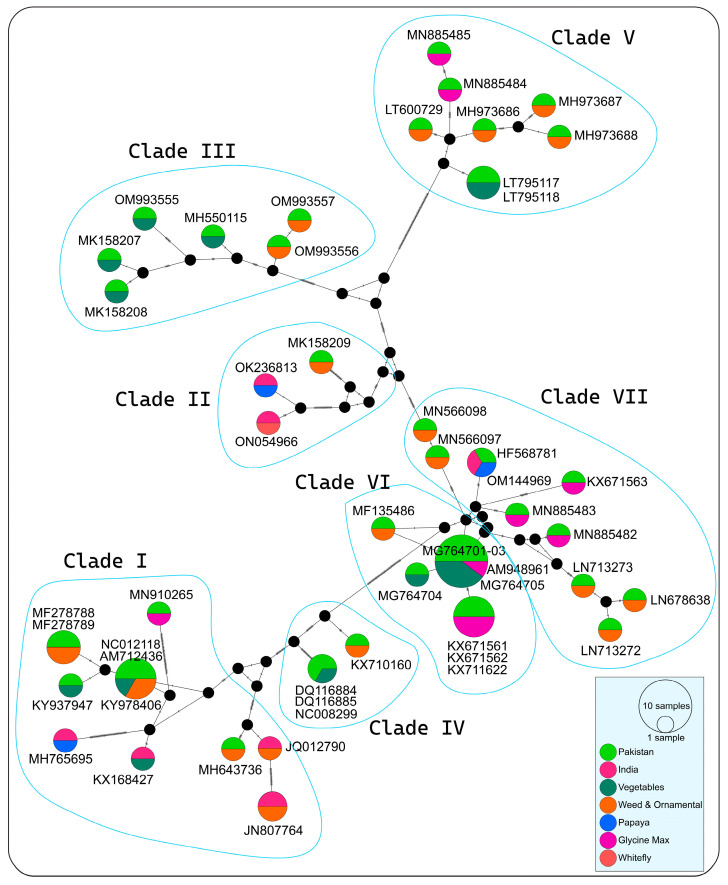
Median-joining network of PeLCV based on full-length sequences constructed in PopART (v1.7). Geographical and host distribution is represented by various colors. Hatching marks indicate the number of mutational steps between sequences. Circle areas are proportional to the number of taxa, and each notch on the links represents a mutated nucleotide position. Missing sequences of PeLCV isolates are indicated by small black circles.

**Figure 3 viruses-15-02358-f003:**
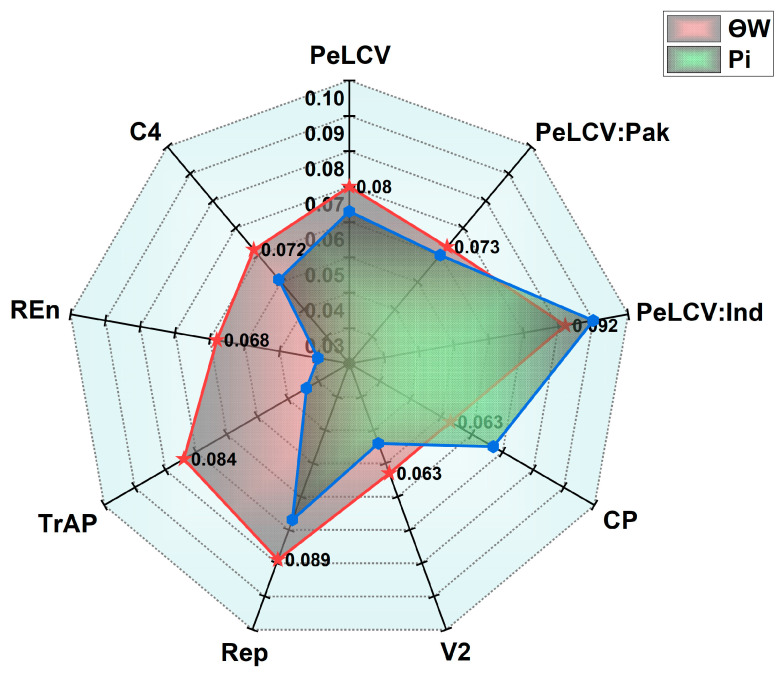
Estimation of genetic diversity (π) and Watterson’s theta (θw) in PeLCV-all, PeLCV-Pak, PeLCV-Ind, and the ORFs encoded by PeLCV.

**Figure 4 viruses-15-02358-f004:**
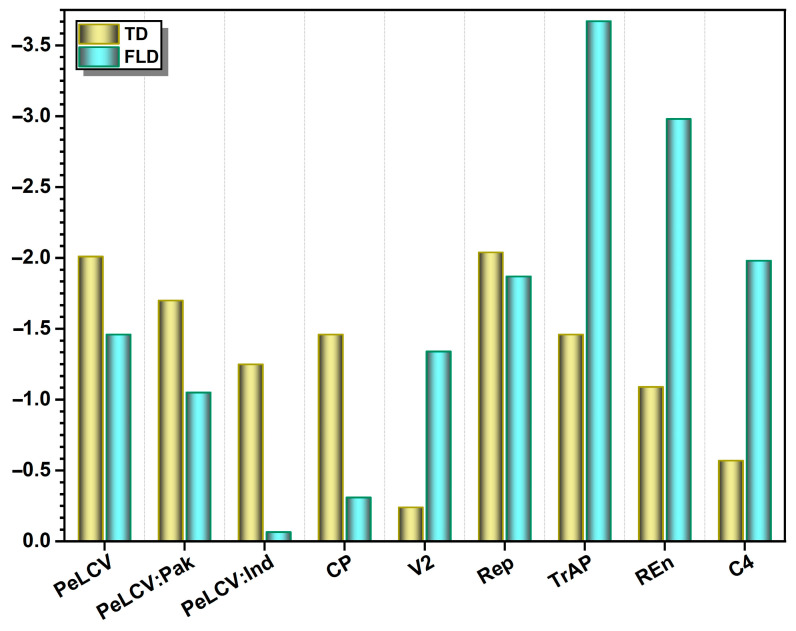
Estimation of Tajima’s D (TD) and Fu and Li’s D (FLD) genetic diversity attributes in PeLCV-all, PeLCV-Pak, PeLCV-Ind, and the ORFs encoded by PeLCV. Abbreviations used are the same as in Table 1.

**Figure 5 viruses-15-02358-f005:**
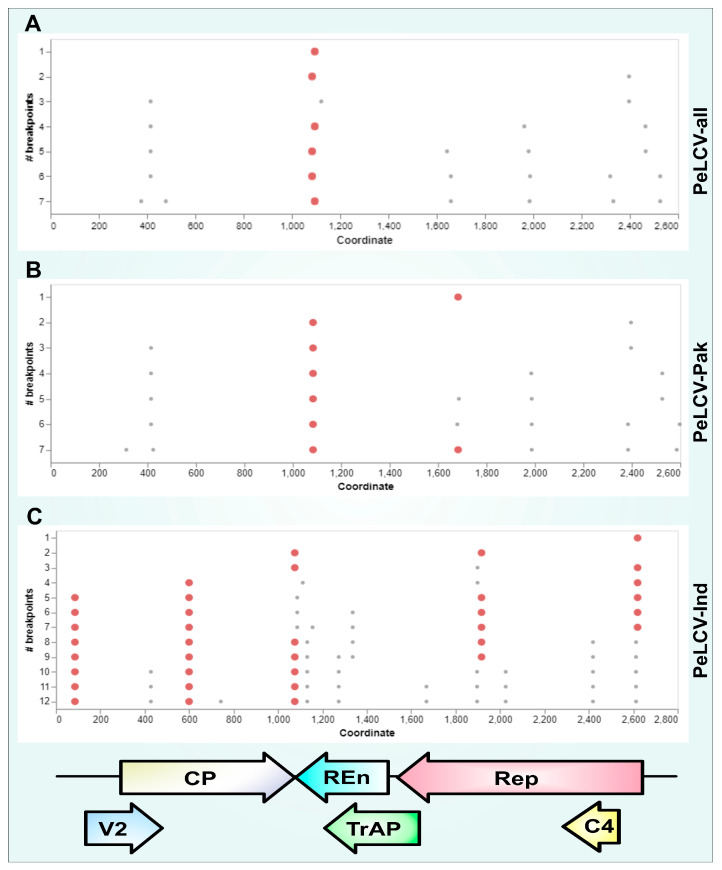
Patterns of recurrent recombination breakpoints distributed within the populations of PeLCV-all (**A**), PeLCV-Pak (**B**), and PeLCV-Ind (**C**). Each bold red dot represents a well-supported recombination breakpoint with a high AICc score. Conversely, each small grey dot represents a breakpoint with a lower AICc score. The linear genome organization of PeLCV, comprising two virion-sense ORFs (V2 and CP) and four complementary-sense ORFs (Rep, TrAP, REn, and C4), is shown at the bottom.

**Figure 6 viruses-15-02358-f006:**
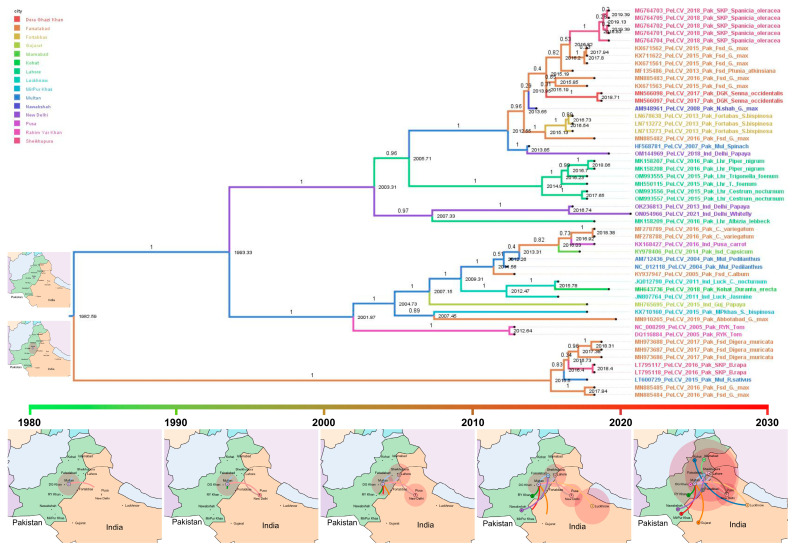
Time-scaled phylogenetic tree of complete genome sequence of all the PeLCV based on Bayesian Markov Chain Monte Carlo method. The estimated time of the MRCA and the names of each PeLCV isolate are marked in the graph, and branches are color-coded based on their respective host types. The findings indicate that the initial split within the PeLCV population occurred around the year 1982, stemming from a common ancestor. The dots represent cities, while the red circle highlights the potential region where PeLCV strain evolved. The colored lines illustrate the movement of PeLCV strains between different regions.

**Table 1 viruses-15-02358-t001:** Genetic diversity indices of PeLCV and its ORFs.

Dataset	No. of Seq	InDel Sites	S	Eta (h)	K	Neutrality Test	
						TD	FLD
PeLCV all	52	86	851	1057	172.08	−2.01	−1.46
PeLCV-Pak	45	152	794	949	175.01	−1.70	−1.05
PeLCV-Ind	07	62	609	684	272.0	−1.25	−0.066
CP	52	2	219	266	59.56	−1.46	−0.31
V2	52	10	92	115	18.38	−0.24	−1.34
Rep	52	18	351	433	67.41	−2.04	−1.87
TrAP	52	24	146	182	16.93	−1.46	−3.67
REn	52	21	118	145	14.97	−1.09	−2.98
C4	52	0	99	118	18.77	−0.57	−1.98

Abbreviations used are insertions and deletions (InDels), total number of polymorphic (segregating) sites (S), total number of mutations (Eta (h)), total number of singleton mutations Eta (S), average number of nucleotide differences between sequences (K), Tajima’s D (TD), Fu and Li’s D test (FLD), coat protein (CP), replication-associated protein (Rep), transcriptional activator protein (TrAP), and replication enhancer protein (REn).

**Table 2 viruses-15-02358-t002:** Estimation of selection pressure on the ORFs encoded by PeLCV.

ORFs	Best Model	Mean Distance (d)	dN	dS	dN/dS	FUBAR Posterior Probability (*p* ≤ 0.9)		SLAC (*p* ≤ 0.05)	
						PS	NS	PS	NS
V2	K2 + G	0.058 ± 0.007	0.034 ± 0.007	0.150 ± 0.028	4.86	8	110	0	8
CP	T92 + G	0.088 ± 0.007	0.049 ± 0.006	0.214 ± 0.025	0.23	2	99	0	50
Rep	HKY + G	0.086 ± 0.008	0.083 ± 0.006	0.061 ± 0.011	1.36	3	112	0	46
TrAP	K2 + G	0.048 ± 0.006	0.035 ± 0.006	0.087 ± 0.016	0.40	3	11	1	10
REn	T92 + G	0.042 ± 0.006	0.028 ± 0.006	0.090 ± 0.018	0.31	4	11	0	8
C4	K2 + G	0.066 ± 0.009	0.044 ± 0.003	0.07 ± 0.010	0.63	9	4	0	3

Abbreviations used in the table are Fast, Unconstrained Bayesian Approximation (FUBAR), single-likelihood ancestor counting (SLAC), non-synonymous (dN), synonymous (dS), general time reversible (GTR), Tamura 3-parameter (T92), Kimura 2-parameter (K2), Gamma distribution (G), coat protein (CP), replication-associated protein (Rep), transcriptional activator protein (TrAP), replication enhancer protein (REn), positively selected sites (PS), and negatively selected sites (NS).

**Table 3 viruses-15-02358-t003:** Mean substitution and codon position mutation rate of PeLCV-all, PeLCV-Pak, and PeLCV-Ind.

Dataset	PeLCV-All		PeLCV-Pak		PeLCV-Ind	
	Strict Clock	Relaxed Clock	Strict Clock	Relaxed Clock	Strict Clock	Relaxed Clock
Mean nt substitution rate (site^−1^ year^−1^)	8.22 × 10^−4^ (225)	8.15 × 10^−4^ (216)	1.44 × 10^−4^ (863)	1.46 × 10^−4^ (539)	8.20 × 10^−4^ (673)	8.13 × 10^−4^ (523)
At 95% HPD interval	8.27 × 10^−4^, 8.19 × 10^−4^	8.16 × 10^−4^, 8.13 × 10^−4^	1.44 × 10^−4^, 1.43 × 10^−4^	1.45 × 10^−4^, 1.44 × 10^−4^	8.26 × 10^−4^, 8.16 × 10^−4^	8.14 × 10^−4^, 8.11 × 10^−4^

Effective sample size (ESS) values are mentioned in parenthesis.

**Table 4 viruses-15-02358-t004:** Codon position mutation rate of the ORFs encoded by PeLCV.

ORFs	Clock Type	Mean nt Substitution Rate (Site^−1^ Year^−1^)	At 95% HPD Interval	CoP1 Mu	CoP2 Mu	CoP3 Mu
V2	Strict	2.39 × 10^−2^ (215)	2.76 × 10^−2^, 2.09 × 10^−2^	1.308 (415)	0.404 (213)	1.289 (520)
	Relaxed	2.48 × 10^−2^ (288)	2.61 × 10^−3^, 2.42 × 10^−3^	1.311 (1899)	0.332 (4381)	1.357 (1488)
CP	Strict	8.03 × 10^−3^ (1508)	9.01 × 10^−3^, 3.44 × 10^−3^	0.397 (4210)	1.086 (6754)	1.52 (7591)
	Relaxed	1.97 × 10^−4^ (1451)	1.76 × 10^−4^, 2.06 × 10^−4^	0.466 (650)	0.928 (541)	1.608 (519)
Rep	Strict	7.1 × 10^−3^ (901)	7.51 × 10^−3^, 6.87 × 10^−3^	0.504 (7237)	1.483 (7734)	1.013 (5767)
	Relaxed	7.21 × 10^−3^ (831)	7.25 × 10^−3^, 7.16 × 10^−4^	0.495 (7669)	1.486 (7998)	1.019 (6349)
TrAP	Strict	2.73 × 10^−4^ (820)	3.71 × 10^−4^, 3.21 × 10^−4^	1.058 (7597)	0.776 (8213)	1.166 (8536)
	Relaxed	2.72 × 10^−4^ (715)	2.79 × 10^−4^, 2.65 × 10^−4^	1.071 (1334)	0.739 (4361)	1.180 (364)
REn	Strict	2.52 × 10^−4^ (712)	2.97 × 10^−4^, 2.17 × 10^−4^	1.085 (7149)	0.612 (6959)	1.302 (7699)
	Relaxed	2.49 × 10^−4^ (661)	2.57 × 10^−4^, 2.43 × 10^−4^	0.978 (7294)	0.636 (7486)	1.386 (6969)
C4	Strict	1.81 × 10^−3^ (881)	2.08 × 10^−3^, 1.72 × 10^−3^	0.271 (7795)	2.105 (6322)	0.624 (6826)
	Relaxed	2.13 × 10^−3^ (505)	2.17 × 10^−3^, 2.09 × 10^−3^	0.282 (2573)	2.093 (2611)	0.625 (2573)

Abbreviations used in the table are codon position mutation (CoP mut) and highest probability density (HPD). Effective sample size (ESS) values are mentioned in parentheses.

## Data Availability

All the data related to this work are available in the manuscript and Appendix A.

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
