# Peer review of "Genetic Diversity, Evolutionary Dynamics, and Ongoing Spread of Pedilanthus Leaf Curl Virus"

_viruses, 2023, doi:10.3390/v15122358_

Round 1
Reviewer 1 Report
Comments and Suggestions for Authors Authors Zafar Iqbal and coworkers presented here a manuscript entitled „Genetic diversity, Evolutionary Dynamics, and Ongoing spread of Pedilanthus Leaf Curl Virus“.The main contribution of the paper lies in the detailed analysis of the evolution of the Pedilanthus leaf curl virus (PeLCV) from the genus Begomovirus. The authors analysed 52 full-length sequences of PeLCV from Pakistan and India. They performed a detailed analysis of evolutionary relationships within PLCV populations from multiple perspectives, using phylogenetic, phylogeographic and recombinant analyses. Their work is highly meritorious because this virus has not yet been studied phylogenetically at the level of populations in different countries, nor has a proper analysis of recombinant phenomena in the virus population been performed. The paper is well written and the methods and procedures used are well described. The results are presented in great detail. The phylogenetic analysis is well performed. The recombination analysis has contributed to the elucidation of the origin of the virus. It is very detailed and all the details are given in the supplementary materials. In the discussion, the authors have properly compared their results with corresponding work published on other viruses. I have no further comments on the work presented.
The paper is worthy of publication in the journal Viruses.
Author Response
The authors express gratitude to the reviewer for investing his time in evaluating our manuscript. We also extend our thanks for promptly recommending the script for publication.
Reviewer 2 Report
Comments and Suggestions for Authors
The paper provides interesting information on the evolution process of Pedilanthus leaf curl virus, a relevant virus in the specific area of Asia. The results conclude that many factors influence the generation of ‘seven distinct clades, displaying regional delineation but lacking any host specific demarcation’. The larger number of sequences analyzed is assumed to be the reason of the additional identified clades .
Nonetheless, no mention is done about the vector of the virus, a species complex (Bemisia tabaci ) that comprises a large number of genetically variable populations, some of which are referred as biotypes.
As well some information on the plant species hosting the virus sequenced, likely different, would be more useful to support the conclusion.
Some comments on the above aspects will make the paper more complete and offer some information of practical importance also for the management of the diseases induced by PeLCV.
